# A Novel Technique for Ultrathin Inhomogeneous Dielectric Powder Layer Sensing Using a W-Band Metasurface

**DOI:** 10.3390/s23020842

**Published:** 2023-01-11

**Authors:** Zachary Kurland, Thomas Goyette, Andrew Gatesman

**Affiliations:** Submillimeter-wave Technology Laboratory, University of Massachusetts Lowell, Lowell, MA 01854, USA

**Keywords:** metasurface, materials characterization, sensing, microwave sensors, powders, materials science, millimeter wave devices

## Abstract

A novel technique using a W-band metasurface for the purpose of transmissive fine powder layer sensing is presented. The proposed technique may allow for the detection, identification, and characterization of inhomogeneous ultrafine powder layers which are effectively hundreds of times thinner than the incident wavelengths used to sense them. Such a technique may be useful during personnel screening processes (i.e., at an airport) and in industrial manufacturing environments where early detection and quantization of harmful airborne particulates can be a matter of security or safety. The proposed sensing technique was experimentally and theoretically tested. The results suggest that, using this technique, one may identify, extract the effective complex dielectric properties, and measure the layer thicknesses of ultrafine powder layers present on a metasurface. Using this technique, it may be possible to identify and characterize diverse media in various physical, chemical, and biological metasurface sensing efforts at numerous bands of the electromagnetic spectrum.

## 1. Introduction

Electromagnetic metasurfaces are the two-dimensional counterparts of metamaterials, which are superficial objects that display exotic properties uncommon or not realizable in natural materials and processes. The success at eliciting a desirable response from an impinging electromagnetic field comes from the careful engineering of sub-wavelength structures and the materials from which they are constructed. Each subwavelength structure consists of three major components: the resonator, the substrate, and the ground (the latter of which may not be present in transmission applications). Together, these three structures define an object called a unit cell. In many ways, a unit cell can effectively be thought of as a voxel. Each of these components interact, producing a local optical response which can be defined by phase, amplitude, and polarization. In periodically arranged metamaterials, one may start by designing a unit cell, and then repeat the same pattern to produce a larger device composed of many of these unit cells. However, similar to pixels on a screen, each element can be identical, or bear unique characteristics which are specifically designed and configured to produce a macroscopic effect that is only apparent when viewing their combined interactions. Indeed, full wavefront control can be obtained by the proper engineering of these unit cells [1].

Over the past two decades, the number of metasurface designs and applications has grown dramatically. Novel devices have been designed and built to function in bands of the electromagnetic spectrum ranging from the microwave to optical and have been applied in military, medical, and fundamental scientific research efforts. To name a few, metasurfaces have improved the function of antennas, created “perfect” superlenses, produced cloaking devices and reduced radar cross-sections, aided in energy harvest from the sun, and have enabled various material identification and detection techniques [2]. Recently, metasurface technology has been used for the sensitive detection of sugars [3], pesticides [4], bacteria [5], viruses [6], antibiotics [7] and many other sensing applications using frequencies typically greater than 1 THz. However, there seems to be a lack of literature utilizing metasurface technology for the purpose of material sensing and characterization at low THz frequencies, such as W-band.

THz waves have proven to be particularly useful in biosensing technology as they are non-ionizing and particularly sensitive to the presence of water. Moreover, THz can penetrate materials such as paper, plastic, and different textile materials which are opaque to visible and NIR waves. Still, attempting to sense samples whose thicknesses are much smaller than a wavelength remains challenging [8]. Therefore, it makes sense that metasurfaces could be used to increase the sensitivity of THz devices since metasurfaces have been shown to amplify light-matter interactions, making electromagnetic field responses much more pronounced [9]. The pronounced light-matter interactions produced by metasurfaces at W-band would be especially useful in industrial manufacturing or mining environments where the sensing of thin layers of dielectric substances is of global interest [10]. In the screening of personnel, it could be that trace amounts of substances could be found on an individual or an unidentified powder is detected. In this paper, a metasurface capable of distinguishing between various powder substances is designed and experimentally tested. The metasurface, as depicted in Figure 1, sits atop a 4-inch diameter, 400 μm thick silicon wafer (dark blue), and the pattern is etched into a 250 nm thick layer of copper (orange). The geometric parameters of the metasurface device are presented in Table 1.

## 2. Materials and Methods

When a metasurface allows for the coupling of the electric and magnetic components of an incident electromagnetic wave to its surface (parallel to their respective incident fields), the local electromagnetic field is greatly amplified [11]. Such a coupling creates electromagnetic “hotspots” within some volume surrounding the device apertures, and a closer free-space impedance match is realized. Furthermore, electromagnetic hotspots greatly enhance the sensitivity of metasurface devices to changes in the refractive index of their ambient environment [12,13]. 

A metasurface device was sought which displayed a strong magnetic dipole moment and electric field vectors parallel to their respective incident fields (indicating coupling). A patch array-like geometry was considered as it allows the metasurface transmittance spectra to be modeled as a modified capacitive patch array using an equivalent circuit model, which will later be used to extract effective dielectric properties and thicknesses of thin powder layer layers on its surface. Such a capability could aid in the identification of such powder layers. The capacitive patch array has been extensively studied and modeled as an equivalent circuit, where strong electric field coupling to aperture slits is observed along the polarization axis of incident electromagnetic radiation [14,15]. The exact methodology for how we arrived at the final metasurface topology is offered in Section 2.1.

### 2.1. Perturbations to Fundamental Resonant Topologies

It will be shown here how by strategic perturbations to fundamental resonant structures (such as a capacitive patch array), novel resonant phenomena can be engineered for particular purposes. Beginning with an example of a simple capacitive patch array, the strong resonant coupling of the electric field to the array was simulated using the commercially available electromagnetic simulation software HFSS and is presented in Figure 2 (four unit cells of the capacitive patch array are shown so that readers may observe vectors which extend beyond a single unit cell). Each patch was considered as copper sitting atop a 400 μm silicon (ϵ=11.9) wafer and was 400 μm × 400 μm in size. The slit widths between each patch were 20 μm. It is seen that strong electric field vectors are produced between the slits of the capacitive patch array and parallel to the incident electric field.

However, since a simple capacitive patch array does not also facilitate the coupling of an incident magnetic field to its surface, perturbations were made to the patches to induce such a coupling (a behavior highly desirable for sensing applications). Evidence of the nonexistence of a magnetic field coupling to a capacitive patch array is presented in Figure 3.

In order to understand how one may modify a simple topology such as that of the capacitive patch array to induce incident magnetic field coupling, it is useful to view the surface currents present on each patch, as depicted in Figure 4. In Figure 4, it is seen that for a simple capacitive patch array, the current vectors flow in a direction parallel to the incident electric field. Therefore, if one makes strategic perturbations to the surface current, a curl can be induced in the current’s field structure. As seen in Figure 5, the perturbations introduced to the topology are slots at the top and bottom of each patch that run perpendicular to the current vectors shown in Figure 4.

Due to the perturbations, the surface current density increases by nearly a factor of five at its greatest and causes the current vectors to curl around the perturbative slots. Therefore, if one imagines the magnetic fields produced due to this new surface current structure seen in Figure 5, using the right-hand rule, it is easily observed that a magnetic field should go into the page at the topmost perturbative slot on each patch, and out of the page at the bottom most perturbative slot of each patch. In fact, this is what is seen when the magnetic fields are observed in the HFSS simulation, as depicted in Figure 6. As can be seen, there is a very strong magnetic dipole induced, extending between the perturbative slots of each patch. There was a thirty-fold increase in maximum magnetic field strength induced directly above the perturbed capacitive patches in comparison with the unperturbed ones (indicating a resonant coupling of incident radiation), making this a viable structure for an electromagnetic hotspot to exist [12,13].

It should be noted that the perturbative slot length and width were chosen arbitrarily for this example (the actual geometries of the final metasurface topology utilized in this study were altered and optimized, and the final form is depicted in Figure 1). It should also be noted that the slots through the horizontal symmetry in the optimized metasurface features (Figure 1) that are perpendicular to the analogous perturbative slots in the patches were made in order to increase the surface current density around each perturbative slot (this is evidenced by an even larger increase in magnetic field strength at each perturbative slot in the optimized topology, as seen in the next subsection).

### 2.2. The Optimized Metasurface Topology

Since it was shown in the previous subsection how one may perturb a simple topology to induce a strong resonant coupling of the incident electromagnetic field to its surface, the design parameters were optimized for our sensing application. A checkerboard array lattice was utilized as it facilitates desirable transmittance spectra for sensing applications (which are shown later in this subsection). The exact metasurface unit cell utilized in this this study is depicted in Figure 1. The electric and magnetic fields of our optimized unit cell, as depicted in Figure 7, show that it facilitates a strong resonant coupling of both the electric and magnetic fields to the topology. Figure 7 focuses on a single aperture from the total unit cell presented in Figure 1 in order to accentuate the field structure for the reader.

As can be seen in Figure 7b, the electric field vectors resemble those seen in an ordinary capacitive patch array (see Figure 2), and there is a strong magnetic dipole moment along the axis of the incident magnetic field (Figure 7d) between each perturbative slot. A magnetic dipole was also observed in Figure 6 (the simply perturbed capacitive patch array), but it is obvious that the induced magnetic field strength on the optimized topology far exceeds even the simply perturbed capacitive patch array. It is also important to once again note the horizontal perturbative slot introduced into the optimized metasurface topology. This slot runs parallel to the incident electric field and through the center of the patch, perpendicular to the two perturbative slots that were introduced (seen in Figure 5 and Figure 6). This horizontal slot was introduced because it drastically increases the current density around each magnetic pole, evidenced by the significant increase in magnetic field at the poles, as seen in Figure 7c,d.

The device made up of many of the unit cells depicted Figure 1 was simulated in HFSS from 90 to 110 GHz (3.33–2.73 mm wavelength). It possesses a resonant frequency of 106.84 GHz and allows 92.6% of the incident radiation to transmit through its surface at resonance. The finalized unit cell parameters, presented in Table 1, are those which produce the greatest simulated maximum transmittance amplitude in the operational bandwidth in which this study was conducted. The transmittance spectrum of the metasurface is presented in Figure 8. The total aperture area of the metasurface makes up approximately eight percent of the total; such a large transmittance amplitude indicates extraordinary transmission has taken place since this observation contradicts Bethe’s predictions [16]. 

The distinct resonant peak observed in Figure 8 is an important feature of the transmittance spectra of the metasurface device to be used in the sensing of fine powders. By considering the maximum transmittance amplitude and resonant frequency of the metasurface device, one may look for the relative shift in maximum transmittance amplitude and resonant frequency of the metasurface when there are fine powder layers on its surface. In doing so, information was gathered which permitted the detection, identification, and characterization of the powder layers which were deposited thereon (the exact methodology and results are presented later).

### 2.3. Metasurface Manufacturing and Characterization

The metasurface was fabricated at the University of Massachusetts Lowell, Emerging Technology and Innovation Center (ETIC) Nanofabrication Laboratory on a 400 μm silicon wafer, which is depicted in Figure 9. In Figure 9, orange represents a 250 nm layer of copper, and the dark aperture-slits are areas without metal to let light pass through the silicon wafer below. The aperture dimensions were very close to those presented Figure 1, and the accuracy and precision of the manufacturing process is best represented in the comparison of the theoretical (HFSS) transmittance spectra and the experimentally acquired spectra, which are be presented later in this subsection.

The transmittance spectra of the bare metasurface device were collected using a Keysight Professional Network Analyzer (PNA) and a pair of W-band horn/lens antennas. A custom apparatus was constructed in order to facilitate measurements of the metasurface and is shown in Figure 10.

The PNA was calibrated using a response calibration technique. Only transmittance data, denoted in this study as $S_{21}$ (the total transmitted power detected by Horn 2 as it leaves Horn 1), was considered. The horns were erected approximately 80 cm apart. The Gaussian beam produced through the focusing lenses at each horn had a FWHM of approximately two inches in diameter in the focal plane and was centered on the metasurface (4-inch diameter). The measurement window was isolated from the ambient environment and the effects of noise were removed from the calibrated data by applying time-domain gating of the space outside of the sample area. Furthermore, the optical bench that the apparatus was erected on, and various metallic surfaces in the area, were covered in mm-wave absorbing material to further reduce error in the measurements. 

The measurements of the metasurface’s operational bandwidth ranged from 90 GHz to 110 GHz with a frequency resolution of 22 MHz and 80 dB of dynamic range. A comparison of the measured and simulated HFSS transmittance spectra through the bare metasurface device is presented in Figure 11. The resonant frequency and transmittance amplitude at resonance are presented in Table 2. The mean absolute error between the experimental and theoretical spectra was 1.33%.

The difference between the experimental and simulated spectra (as seen in Figure 11) may be attributed to an inexact match between the metasurface slit dimensions, which are highly dependent on the many factors involved in manufacturing, such as photo-resist thickness, UV flood exposure time, development time, cleanliness of the substrate and undercut severity [17,18,19,20]. Furthermore, although the agreement in bare antenna transmittance spectra between experiment and simulation is good, their differences in maximum transmittance amplitude and resonant frequency (Table 2) must be kept in mind when comparing the experimental and simulated powder spectra in later sections as the differences (specifically in maximum transmittance amplitude) are physically significant on the scale at which the powder spectra were acquired.

### 2.4. Fine Powder Layer Measurement Using a Metasurface

In order to study the effect that fine powders have on the resonant properties of the metasurface device and fully characterize them, measurements were made experimentally and simulated using HFSS. Experimental measurements of four powders were conducted in two separate runs, two weeks apart. The four powders studied using the metasurface device were baby powder, baking powder, baking soda, and calcium carbonate. These powders were chosen because they were readily available and may be easily acquired. Although these powders are benign in nature, they are suitable for showing the strength of the sensing technique presented herein as they appear physically similar upon visual inspection and may not be easily distinguished. We propose that the sensing technique presented in this paper may be applied to many other types of powders of interest which may pose a risk to the health and safety of others. 

The four powders were independently deposited onto the metasurface (as it sat on the apparatus, as seen in Figure 10) in 200 mg increments, up to 1000 mg through 45–65 μm mesh sifters, held roughly one-inch above the surface of the metasurface. After each 200 mg layer was deposited onto the metasurface, transmittance spectra of the powder coated metasurface was acquired from 90 to 110 GHz. The frequency resolution of these measurements was about 22 MHz (901 frequency steps total). The powders were distributed across the surface of the device as evenly as possible, but it must be noted that upon deposition onto the surface, some fraction of the sifted powder would become airborne and not land on the metasurface. The total amount of mass left on the metasurface after attempting to sift 1000 mg onto its surface (in 200 mg increments) is presented in Table 3. We waited to measure the powder mass on the metasurface after 1000 mg was attempted to be deposited onto it so as not to disturb the apparatus or metasurface position between successive layer measurements.

Upon sifting, some powder particulates clumped together on the surface of the device, and larger particulates occasionally created small craters in the fine powder layers, making them inherently inhomogeneous. Furthermore, it is worth noting the different masses between the two experiment runs presented in Table 3. Without exception, the second run of the experiment yielded final masses on the metasurface which were greater than the first. It is suggested that the hygroscopic properties and fluctuations in the relative humidity of the storage environment may have led to this difference between experimental runs. In one study, some hygroscopic powders adsorbed up to twenty percent of their mass in water vapor from the ambient environment with improper storage [21]. It also was found that in an ambient environment of twenty-five to seventy-five percent relative humidity, some powders adsorb up to six percent of their mass in water during routine handling. Therefore, due to the hygroscopic nature of the powders, the two-week period between measurements may have changed the effective masses of the powder particulates, making them less susceptible to becoming airborne and float away from the metasurface [21,22]. Furthermore, we acknowledge that the relative humidity is a factor to be considered in the resonant response of the metasurface to thin powder layers, as the relative permittivity of water at 100 GHz is approximately $\epsilon_{H_{2}O}=8+i15$ [23]. 

### 2.5. Bulk Dielectric Properties of Powders and Maxwell Garnett Effective Medium Theory

We sought to establish a set of controls for the powders tested in this study by simulating the resonant response of the metasurface to the thin, inhomogeneous powder layers on its surface in HFSS. The effective complex dielectric properties of the powders were defined by the well-established Maxwell Garnett effective medium theory, whose layer thicknesses were defined by an average effective thickness (derived experimentally and shown below). The effective complex dielectric constants and thicknesses of each powder layer are the only pieces of information necessary to simulate the resonant response of the metasurface to the thin powder layers on its surface in HFSS. 

The effective permittivity of an inhomogeneous layer was defined by the Maxwell Garnett effective medium theory, using the following equation:(1)$$\epsilon_{eff}=\epsilon_{m}\frac{2\;\delta_{i}\left(\epsilon_{i}-\epsilon_{m}\right)+\epsilon_{i}+2\epsilon_{m}}{2\epsilon_{m}+\epsilon_{i}-\delta_{i}\left(\epsilon_{i}-\epsilon_{m}\right)}\;,\;$$
where $\epsilon_{m}$ is the component of the bulk complex permittivity of the matrix in which the inclusions, whose permittivities are $\epsilon_{i}$, are set, and $\delta_{i}$ is the fill factor of each powder layer (the volume fraction of inclusions, between zero and one). In this study, the inclusions were considered to be air, whose permittivity is $\epsilon_{air}=1+i0\;$, and the bulk matrix, the powder in question. To obtain the bulk dielectric properties of the powders, $\epsilon_{m}$ (to be used in the Maxwell Garnett effective medium theory), they were individually compressed into a 3-inch diameter disk using a 15-ton hydraulic press, and their reflectance spectra were measured using the apparatus presented in Figure 10. The bulk complex dielectric constants were obtained through Fresnel fitting of the reflectance spectra, whose values are presented in Table 4.

We extracted the average effective thicknesses of each 200 mg layer deposited onto the metasurface using the equation:(2)$$\;T_{ave}=\frac{\sigma_{p}}{5\cdot \rho_{p}}.$$

In Equation (2), $\sigma_{p}$ is the “average surface density” of each powder on the metasurface and $\rho_{p}$ is the volumetric density of the powder as it was deposited onto the metasurface. The average surface density is the total amount of powder left on the metasurface after attempting to deposit 1000 mg onto it, divided by the area of the silicon wafer. The volumetric density of each powder as it rested on the metasurface was estimated separately by sifting each powder into a 0.125-inch thick, 2-inch diameter cylindrical plastic container, measuring the mass of the powder within, and dividing by the volume of the cylindrical plastic container. This method for estimating $\;T_{ave}$ was utilized because it was not possible to directly measure the thickness of the powder after it had been deposited onto the metasurface. The factor of 5 in Equation (2) is present because there are five 200 mg increments which make up the 1000 mg total. The average effective thickness of each powder layer is presented in Table 5.

Then, the sifted fill factors, $\delta_{i}$, which are the average fractional amount of air present in each 200 mg layer of powder as they had been sifted onto the metasurface were determined using the equation:(3)$$\delta_{i}=1-\frac{\rho_{p}}{D_{p}}.$$

In Equation (3), $\rho_{p}$ is the volumetric density of the powder as it was deposited onto the metasurface (as seen in Equation (2)), and $D_{p}$ is the density of the hydraulically pressed powder. The values of $\delta_{i}$ are presented in Table 6.

The thin powder layers were simulated in HFSS as dielectric clads which filled each aperture of the metasurface and extended above its surface, possessing the average effective thicknesses (Table 5) and complex dielectric properties obtained using Maxwell Garnett effective medium theory. The dielectric properties obtained by use of the Maxwell Garnett effective medium theory are presented in Table 7.

The results of the HFSS simulations are presented in Figure 12. In Figure 12, it is seen that each of the resonant transmittance values which belong to an individual powder lie relatively far from one another. However, we suggest that one should not consider single data points (resonant transmittance values produced by a single powder layer) alone, since the relative shifts in transmittance amplitude between different powders is small. At the very least, when considering the entirety of each of the spectra produced by each powder (five data points corresponding to the five powder layers tested), it is clear that they are distinguishable from one another in theory. This provided some confidence that we may be able to distinguish individual powders from one another using this technique, and we therefore, sought experimental verification of our suspicions (which were validated, as will be shown in the next section).

## 3. Results

### 3.1. Experimentally Acquired Transmittance Spectra of Fine Powder Layers

The transmittance spectra of the four powders acquired over the two experimental runs (conducted two weeks apart) were gathered using the apparatus shown in Figure 10 and are presented in Figure 13. Looking at Figure 13, each point represents the maximum transmittance amplitude and the corresponding resonant frequency of each of the acquired transmittance spectra. It is clear that there are distinct differences in the evolution of the transmittance spectra between the four powders tested as a function of layer thickness. However, important similarities are observed between alike powders between successive runs, demonstrating the reproducibility and distinguishability of the proposed sensing technique. Furthermore, although we acknowledge that the relative shifts in transmittance amplitude between successive powder layers are relatively small, we suggest that this technique shows that the scale of the relative shifts in resonant values are not of great importance. We propose one must consider the resonant transmittance values of more than one powder layer on the metasurface before attempting to identify a particular powder. In so doing, one may consider each of the data points as an individual member of a distinct resonant spectral evolution belonging to a specific powder. 

In this way, by viewing the evolution of the resonant transmittance values of a particular powder as a function of layer thickness, a distinct spectral signature may be produced. This spectral signature may allow one to identify a particular powder on the metasurface since, as seen in Figure 13, no two powder species share even a similar spectral signature. Furthermore, we suggest that the spectral extremum of each powder could be acquired as a function of humidity in further experimentation, thus constraining the possible spectral evolutions for each powder. Doing so could produce a set of possible spectral signatures that may be observed using this sensing technique.

It is clear that there is a striking difference between the spectra simulated in HFSS using Maxwell Garnett effective medium theory (Figure 12) and the experimental spectra (Figure 13). However, they do agree with one another in that a decrease in relative resonant frequency shifts between successive powder layers is observed as the layer thickness increases (with the exception of baking soda). In further agreement with theory, the greatest relative shift in resonant frequency between adjacent resonant points of the same species is observed when increasing the layer thickness from 0 mg to 200 mg, as was observed in the experiments, indicating that the device is sensitive to minute changes in the ambient environment.

In comparing the two experimental runs, it is clear that there are similarities and differences between identical powders, but there are still unique characteristics belonging only to powders of the same species. For instance, calcium carbonate and baking soda both reliably produced transmittance spectra which rose above the bare antenna resonance values. However, spectra produced in HFSS using Maxwell Garnett effective medium theory and average effective thicknesses never produced a transmittance amplitude that surpassed the bare antenna peak. Furthermore, baking soda produced an interesting result in the experiments: in Run 1 there is a large jump in resonant frequency and transmittance amplitude from the 200 mg layer to the 400 mg layer and a similarly perplexing jump from the 600 mg layer to the 800 mg layer in Run 2 (diverging from the pattern observed in simulation and distinct from the other powders’ spectral evolution observed experimentally). 

Since the two runs were carried out two weeks apart and the same anomalous transmittance responses were observed each time in baking soda and calcium carbonate (while baby powder and baking powder also remained consistent in their respective spectral evolutions between experimental runs), the anomalous responses may not be easily written off. These “jumps” and transmittance amplitudes which extend above the bare metasurface response may offer insights into some key physics taking place between the powders and the metasurface apertures (a possible explanation detailing the physical significance of these anomalous observations is offered in Section 4). 

Looking at Table 3, the difference in total powder mass on the metasurface suggests that the spectral signatures of similar powders between experimental runs should not be identical, and indeed, they differ slightly from one another (save calcium carbonate). However, the discrepancies between simulation using Maxwell Garnett effective medium theory and the experimentally observed anomalous behavior of the thin powder layers are too great to ignore. These anomalies suggest that perhaps a new effective medium theory be realized in order to fully characterize the complex dielectric properties and effective thicknesses of thin dielectric powder layers. 

### 3.2. An Equivalent Circuit Model of a Metasurface

This section serves as a detailed explanation of the techniques employed to produce an equivalent circuit model which reproduced the transmittance spectra acquired through simulation with a high degree of accuracy. It will be later shown in Section 3.4 that by using the equivalent circuit model produced in this subsection to extract complex dielectric properties and effective layer thicknesses of the thin powder layers studied herein (to be used as parameters in HFSS simulations), better agreement between experimental and simulated spectra is obtained. In doing so, the physical properties of the powders were more clearly illuminated.

The series RLC circuit was the circuit upon which all transmittance spectra presented in this paper were described as each spectrum is singly resonant in nature [14]. For our metasurface topology, it was assumed that it behaves similar to a capacitive patch array as the electric field of the metasurface exhibits similar behavior between the gaps orthogonal to the incident electric field (as seen in Figure 7a,b), and it is patch-like in shape. The following subsections detail the process by which the values of *R*, *L*, and *C* were determined for the bare metasurface (to be used in the equivalent circuit method to determine powder properties).

#### 3.2.1. The Capacitance (*C*) of the Metasurface

Since the metasurface is doubly periodic and, therefore, contains a non-trivial inter-patch spacing, the equation for the capacitor of the sought equivalent RLC circuit model was slightly modified from the classic form [14] to account for the double periodicity and variable inter-patch spacing, the form of which is presented in Equation (4) [24,25].
(4)$$C=\frac{P\epsilon_{0}\left(\epsilon_{r_{1}}+\epsilon_{r_{2}}\right)}{\pi }\ln\frac{1}{\sin\frac{\pi g}{2P}}$$

In Equation (4), g is the average distance between two active components within the unit cell, *P* is the lattice period, $\epsilon_{r_{1}}$ and $\epsilon_{r_{2}}$ are the effective permittivities of the substrate and the material directly above it, respectively, and $\epsilon_{0}$ is the permittivity of free space. The value of the capacitance of the RLC circuit is an essential parameter, for it is solely responsible for changing the position of the resonant peak of transmittance spectra between active circuit ports in frequency alone. 

##### Finding the Average Inter-Patch Spacing (g)

It was determined that the average inter-patch spacing between the two active components of the unit cell, g, could be easily found by utilizing Equation (5).
(5)$$\;\;\;g=\frac{\int_{0}^{\frac{L}{2}}\int_{0}^{x}\sqrt{1+{\left(\frac{dy}{dx{}^{\prime }}\right)}^{2}}dx{}^{\prime }dx}{\int_{0}^{\frac{L}{2}}dx}\;$$

In Equation (5), $dy/dx{}^{\prime }$ is the slope of the solid black line labeled $dy/dx{}^{\prime }$ in Figure 14. Since $dy/dx{}^{\prime }$ = 1, after integration of Equation (5), it is found that g = $\frac{L}{2\sqrt{2}}=287\;\mu m$, where *L* is depicted in Figure 1.

##### Finding the Metasurface Period (*P*)

Since the metasurface is doubly periodic and, therefore, is non-trivial, the Brillouin Zone of the metasurface lattice is utilized to find *P*, which is ultimately defined by the fact that the metasurface possesses an orthorhombic unit cell. Using the fact that the lattice is orthorhombic, the resonant wavelength of the two-dimensional metasurface lattice is defined as the dispersion relation for an orthorhombic lattice generalized to two dimensions [26,27]. Doing so produces Equation (6), which provides the relationship between the resonant wavelength of an orthorhombic lattice and its period, *P*.
(6)$$\lambda_{o}=\frac{P}{\sqrt{\frac{m^{2}}{a^{2}}+\frac{n^{2}}{b^{2}}}}\sqrt{\frac{\epsilon_{d}\epsilon_{m}}{\epsilon_{d}+\epsilon_{m}}}\;\;$$

In Equation (6), $\epsilon_{m}$ is the relative permittivity of the metal, $\epsilon_{d}$ is the relative permittivity of the dielectric in contact with the metal (the substrate), and (*m, n*) are the grating orders. *P* is the period of the array, *a* and *b* are the Miller Indices [27], which are (*a, b*) = (2, 1) for our device. If the leading diffraction order is chosen as (*m, n*) = (1, 0), with the limit $\epsilon_{m}\gg \;\epsilon_{d}$, the period of the array is found by solving Equation (7) [26,28,29].
(7)$$P=\frac{\lambda_{o}}{2\sqrt{\epsilon_{d}}}\;\;$$

In Equation (7), if $\lambda_{o}$ is the resonant frequency of the metasurface device and $\epsilon_{d}=11.9$ (silicon wafer), then P=407 μm, which is 0.25% off the value of $\left|\vec{P}\right|$ in Figure 1. *P* is not random but a length of the interior wall of the Irreducible Brillouin Zone depicted in Figure 15 as $b_{i}$. The Irreducible Brillouin Zone represents the smallest area on the lattice surface which completely characterizes a fundamental cell as knowing the solutions to the wave equation at each point within it also defines the solution everywhere on the reciprocal lattice [30].

##### Solving for the Capacitance (C)

In the case of the bare metasurface, in Equation (4), $\epsilon_{r_{1}}=11.9,\;\epsilon_{r_{2}}=1$, and it was found that $g=\frac{L}{2\sqrt{2}}$ and P=407 μm. Thus, solving Equation (4), it was found that *C* = 1.64 fF.

#### 3.2.2. The Inductance (*L*) of the Metasurface

Since at resonance the circuit impedance becomes purely resistive, one can derive an expression for the inductor as a function of the resonant frequency and the value of a capacitor [31]. This equation is presented as Equation (8).
(8)$$\;\;L=\frac{1}{\omega_{o}^{2}C\;}\;\;$$

In Equation (8), $\omega_{o}$ is 2π times the resonant frequency of the bare metasurface $f_{o}=106.84\;\mathrm{GHz}$, and *C* = 1.64 fF. This produces an inductor value of L=1.35 nH, which is held constant in all the following circuit simulations presented in this study (why this is so is explained later in Section 3.2.4).

#### 3.2.3. The Resistance (*R*) of the Metasurface

Since the metasurface simulated in HFSS is not perfectly transmissive at resonance, a resistor was placed in series with the inductor and capacitor to accurately model the resonant response of the metasurface. The resistor lowers the amplitude at which resonance occurs and does not affect the resonant frequency. Similarly, holding the value of the inductor constant in this study, the capacitor explicitly controls the position of the resonant frequency. 

To find *R*, the resonant response of the fundamental circuit (the circuit with the constant capacitance and inductance of the bare metasurface) as a function of the resistor value was modeled by utilizing the fact that the envelope of the energy dissipation rate in a series RLC circuit falls off in a manner proportional to $\exp\left(-t\frac{R}{L}\right)$. Upon probing the transmittance response of the equivalent circuit model in QUCS (a circuit simulation software) to changes in *R*, the best fit very closely matches:(9)$$S_{21}\left(R\right)=\frac{2}{3}e^{-t\frac{R}{L}}+\frac{1}{3\;}\;,\;\;$$
where *L* is the equivalent circuit value of the inductor (the metasurface inductance, 1.35 nH), a constant throughout this paper, and $t=f_{o}^{-1}\;$, which is the inverse of the resonant frequency of the bare metasurface. Equation (9) allows one to estimate a value for *R* given any maximum transmittance amplitude through the metasurface–powder interactions. The graph and best fit of the transmittance vs. resistor value for the fundamental circuit, given by Equation (9), is presented in Figure 16. 

Fitting the response of the fundamental circuit to changes in resistor value to Equation (9) produces a good agreement, within 0.5%. The correct series resistor value was determined to be *R* = 7.92 Ω for the bare metasurface transmittance amplitude as simulated in HFSS. 

#### 3.2.4. The Resulting Equivalent Circuit Model for the Bare Metasurface

Since *R*, *L* and *C* for the bare metasurface are known values, the equivalent circuit for the metasurface is presented in Figure 17. It was found that one could change the position of the resonant peak of the transmittance spectra of the equivalent circuit by changing the value of the capacitance and resistance and holding the inductance constant. This fact will prove advantageous in determining effective dielectric constants and layer thicknesses of powders, whose methodology is be presented in Section 3.3.

The transmittance spectra of the circuit were simulated using QUCS. It was shown that the equivalent circuit with the values of *R*, *L*, and *C* derived in the previous subsections produces a unique solution which matches the bare metasurface’s transmittance spectrum obtained in HFSS (Figure 18). It was found that there was a 0.95% difference between the transmittance spectra produced in QUCS and those obtained through metasurface simulation in HFSS. This agreement confirms that our metasurface was justly treated as a modified capacitive patch array.

### 3.3. Effective Complex Dielectric Properties and Thicknesses of Thin Inhomogeneous Dielectric Powder Layers Extracted from an Equivalent Circuit Model of a Metasurface

It was previously shown that poor agreement between the experimental (Figure 13) and simulated (Figure 12) transmittance spectra of thin inhomogeneous powder layers was produced using average effective thicknesses and Maxwell Garnett effective medium theory (whose values were derived using the methods described in Section 2.5). We propose that if one could find a set of complex dielectric constants and powder layer thicknesses which (when simulated in HFSS) produce relative shifts in resonant frequency and transmittance amplitude for each powder layer observed experimentally, that the powder layers could be effectively characterized. After the dielectric properties of each powder layer were extracted using the equivalent circuit approach (and each effective powder layer thickness approximated), simulations of the metasurface with powder layers on it (which carry the equivalent circuit derived effective complex dielectric properties and layer thicknesses) were conducted in HFSS. The resulting simulated resonant spectra were then compared to those obtained via experimentation. The exact method employed to perform this study is detailed directly below.

Since an equivalent circuit model was obtained (Section 3.2.4) which describes the bare metasurface transmittance spectra to within 1% (sharing the same maximum transmittance amplitude and resonant frequency), it was suggested that it could be used to extract the effective complex dielectric properties and layer thicknesses of each inhomogeneous powder layer tested. This is possible in part because the value of the capacitor in the equivalent circuit (see Equation (4)) is determined solely by the permittivity of the material in contact with the metasurface, $\epsilon_{r_{2}}$ (where all the other parameters in Equation (4) are static, physical features of the metasurface structure). 

As a note, since each powder layer has a unique layer thickness and set of complex dielectric constants produced using this technique, for sake of brevity, only Run 2 experimental data will be characterized and presented in this paper (whose resonant spectra are presented in Figure 13).

Though the differences in resonant frequency and transmittance maxima between the two curves in Figure 11 appear to be small, they are significant on the scale of the experimentally measured data (see Figure 13, solid and hollow square data points). In order for the equivalent circuit model to extract accurate complex dielectric constant and effective thickness values suitable for predicting experimental spectra, the differences between measured and simulated resonant frequency and transmittance maxima were compensated for. Consequently, the relative shifts in transmittance amplitude and resonant frequency were used to compare the simulated and experimentally acquired transmittance spectra. Therefore, if a powder produced a resonant frequency and maximum transmittance amplitude in the experiment which departed from the bare antenna resonance values by $df_{o}\;$(%) and $dS_{21}\;$(%), respectively, the resonant frequency used in the acquisition of effective dielectric constants and thickness values using the equivalent circuit technique was:(10)$$f_{r}=\left(1+\frac{df_{o}}{100}\right)f_{o}^{HFSS},\;\;$$
and similarly for transmittance:(11)$$S_{21,r}=\left(1+\frac{dS_{21}}{100}\right)S_{21}^{HFSS},\;\;$$
where $S_{21}^{HFSS}$ and $f_{o}^{HFSS}$ are the transmittance amplitude and resonant frequency of the bare antenna, respectively, as simulated in HFSS. Values of $f_{r}$ and $S_{21,r}$ for each powder layer are presented in Table 8. Since the value of the inductor is held constant (1.35 nH) in this study, one can plug in the shifted resonant frequency values (found after solving Equation (10) for each powder layer) into Equation (8), and the effective value of the capacitor, C, can be found for each powder layer. Then, by plugging *C* into Equation (4), one can solve for $\epsilon_{r_{2}}$, where all other parameters are known. The values of the real component of the effective permittivity, $\epsilon_{r_{2}}$, of each powder layer found using the relative frequency shifts are presented in Table 8 (labeled as $\epsilon_{eff}^{{}^{\prime }})$.

By holding constant two of the three parameters necessary to perform a simulation in HFSS (the thickness and the real component of the complex permittivity), the response of the metasurface to changes in the value of dielectric loss tangent for powders could be studied. The thickness for each powder species was kept constant at the median thickness value of the five powder layers previously estimated using the average effective thickness method, which are presented in Table 5. The median thickness of each of the species, therefore, was three times the individual average effective thicknesses in Table 5 (since in total five layers were applied). The real component of the permittivity of each of the clads were also held constant at the median value extracted by use of the equivalent circuit model, presented in Table 8, for each powder species. Justification for this procedure follows below.

This treatment assumed that the transmittance amplitudes of the spectra should remain relatively stable upon changes to the thickness or real component of the permittivity, which was later confirmed. Upon parametric fitting of the response of the metasurface-powder transmittance spectra to changes in the loss tangent of a powder, a unique equation for each powder as a function of loss tangent was obtained. This allows one to estimate the loss tangent of each powder layer as a function of transmittance amplitude using the equation: (12)$$\;S_{21,r}\left(tan\delta \right)=S_{21,o\;}e^{-\frac{1}{\alpha }\frac{t}{L}R_{min}\frac{S_{21,o\;}}{tan\delta_{exp}}tan\delta }+\beta tan\delta +\gamma .\;\;$$

In Equation (12), α can take the values of 1 or 4 for each powder, and $\frac{t}{L}R_{min}$ is the exponential component which characterizes the energy dissipation of the RLC circuit. $R_{min}$ is the minimum resistor value predicted via Equation (9) for each powder (the minimum resistor value was chosen arbitrarily since fit parameters may account for changes). $S_{21,o\;}$ is the transmittance amplitude of the metasurface as predicted in HFSS with no powder on the surface, $tan\delta_{exp}$ is the loss tangent of the pressed powder whose effective values were extracted using a Fresnel fitting procedure and Maxwell Garnett theory, and β and γ are constant factors which are found using a parametric fitting procedure. 

The justification for using $tan\delta_{exp}$ as an estimate in Equation (12) lies in the fact that the volumetric density and fill factor of the powders (which are directly determined by their densities) as they sat on the metasurface (after it was attempted to apply 1000 mg) were known with greatest certainty. Equation (12) allows one to plug in a relative transmittance value,$\;S_{21,r}$, into the left-hand side, and the appropriate parameters into the right-hand side; then, using a computational solver of choice, a value for the loss tangent of the powder layer which produces the input relative transmittance amplitude was obtained. 

It was determined that even though two of the three parameters necessary to perform a simulation in HFSS were held constant, a much better agreement in transmittance spectra between experiment and simulation was observed in comparison with simulations using Maxwell Garnett effective medium theory and the average effective thickness treatment of each powder layer (still, we felt it could be improved upon even further by extracting effective layer thicknesses with the aid of the equivalent circuit model). The final values of the loss tangent for each powder layer are presented in Table 8 as $tan\delta_{eff}$. A possible explanation of the significance of the negative values of $tan\delta_{eff}$ is offered in Section 4.

Finally, to further increase the agreement between experiment and theory, more accurate effective thicknesses of each powder layer were determined. For each 200 mg layer of powder that was attempted to be deposited onto the metasurface, the percent divergence from the resonant frequencies of the bare antenna resonance (in experiment and HFSS) versus the mass of each powder layer were parametrically fit using a third order polynomial of the form:(13)$$\;f\left(M\right)=\;p_{1}\cdot M^{3}+p_{2}\cdot M^{2}+p_{1}\cdot M+p_{4},\;\;$$
where $p_{i}$ are the parameters of best fit and M is the layer-mass in milligrams. It was found that the adjusted effective thickness ($T_{adj}$) of each powder layer could be estimated using Equation (14).
(14)$$T_{adj}={\left(\frac{f\left(M_{exp}\right)}{f\left(M_{circ}\right)}\;\right)}^{2}\langle \;T_{exp}\;\rangle \;\;$$

In Equation (14), $f\left(M_{exp}\right)$ is the equation of best fit for the experimental relative frequency shifts, $f\left(M_{circ}\right)$ is the equation of best fit for the the relative frequency shifts obtained using the equivalent circuit method previously described, and $\langle \;T_{exp}\;\rangle$ is the median thickness used to simulate each powder while probing the resonant response of the metasurface as a function of loss tangent. The success in utilizing the adjusted effective thickness values with equivalent circuit derived dielectric properties is presented in the next subsection. The improved estimates of effective thicknesses $\left(T_{adj}\right)$ of each powder layer are presented in Table 8.

### 3.4. Simulated Transmittance Spectra of Thin Inhomogeneous Dielectric Powder Layers Using Equivalent Circuit Derived Parameters

We sought an effective medium theory whose dielectric constants and layer thicknesses might produce simulated transmittance spectra (in HFSS) that more accurately represent those observed in experiments (and therefore physically characterize the powder layers) in a way the Maxwell Garnett effective medium theory could not. Such an ability could aid in the identification of powders present on a metasurface. We elected to use an equivalent circuit model to extract the complex dielectric properties and effective layer thicknesses of each powder layer experimentally tested (whose values were found using the methodology of Section 3.3). The spectra produced in HFSS using the three essential components estimated using the equivalent circuit method ($T_{adj}$, $\epsilon_{eff}^{{}^{\prime }}$ and $tan\delta_{eff}$, whose values are presented Table 8) are presented in Figure 19.

As can be seen in Figure 19, the spectral signatures simulated in HFSS (using equivalent circuit derived complex dielectric properties and layer thicknesses) have forms much more familiar than those produced in HFSS using Maxwell Garnett derived complex dielectric constants. This becomes increasingly evident when one directly compares the spectral signatures of Run 2 data (Figure 13) with those presented in Figure 19. However, because of the difference in experimental bare metasurface resonance values and those produced in simulation (see the difference between solid and hollow squares in Figure 12), the “shape” of each spectral signature alone is not sufficient to make a true comparison. Therefore, we offer the relative shifts in frequency and transmittance amplitude from the bare metasurface simulated using the equivalent circuit method derived complex dielectric constants and thicknesses (HFSS) to be compared with the experimentally acquired transmittance spectra and those simulated using the Maxwell Garnett effective medium theory derived complex dielectric constants (HFSS). The relative shifts in resonant frequency and transmittance amplitude induced by the presence of a powder layer (the relative shifts in maximum transmittance amplitude and resonant frequency from the bare metasurface resonance values found in experiment and simulation) are presented in Figure 20 and Figure 21, respectively.

In Figure 20 and Figure 21, there is a marked improvement in agreement between the relative frequency and transmittance amplitude shifts measured in the experiments (in black), those determined using circuit derived complex dielectric constants (in red) and the values of effective thicknesses over Maxwell Garnett effective medium theory (in blue). This is true for all examples presented herein, except for the relative shifts in transmittance amplitude for the 800 mg and 1000 mg baby powder layers (Figure 21a).

It was determined that it is possible to describe the anomalous transmittance enhancement above the bare antenna transmittance value as an effective medium with known thicknesses and possessing negative imaginary components of the effective permittivity. This suggests that there may be some enhancement mechanism which is not accounted for using the Maxwell Garnett effective medium theory (as it was observed in both experimental runs). A possible explanation for this enhancement is provided in Section 4). However, because the proposed equivalent circuit method with which the effective thicknesses were extracted and the complex dielectric constant values derived utilizes equations of best fit for the loss tangent, it is clear that a complete theory needs to be realized.

## 4. Discussion

The following is offered as a physical explanation for the observed disagreement between results obtained by use of Maxwell Garnett effective medium theory and experiments.

Effective medium approximations (such as Maxwell Garnett or Bruggeman) assume that the effective complex dielectric properties remain the same between successive powder layers as they stack upon one another. They also require that the local electric field fluctuations in inhomogeneous media be of the order of the average field (the inhomogeneities on the surface and within a media are referred to as defects). In particular, if the defects are smaller than, or much smaller in diameter than one-quarter of the incident wavelength, finite-size effects of the media do not contribute to fluctuations in the electric field intensity at the local scale, requiring these defects be treated as distinct features [32].

At the local scale, defects can greatly enhance and focus electric fields, resulting in significant deviations in electric field intensities. It was shown by Risser and Ferris that the polarization of these defects rapidly and nonlinearly increases until the lattice is approximately sixty-five percent filled with defects, at which point saturation is reached. At this juncture, the medium assumes a linear behavior in its polarization change as a function of fill factor. 

The average fill factor for each layer never exceeded sixty-three percent in this study, indicating that the polarization of the defect sites may have changed in a nonlinear way between the applied layers. A nonlinear increase in polarization strength as a function of fill factor produces a widened distribution of electric field values locally within an inhomogeneous media and is not accounted for in classical or modified effective medium approximation methods. This suggests that no known effective medium theory is suitable to account for such local field enhancements [33].

The defect model may explain why the proposed method to extract the complex dielectric constants and thicknesses of powders from an equivalent circuit model is much more successful at predicting the powder-metasurface resonant responses than effective medium theories such as Maxwell Garnett (which treat each layer of the powder as possessing identical complex dielectric constants). This is because a key feature of the technique described in this paper is to assign each powder layer a complex dielectric constant which changes as a function of thickness. Furthermore, regardless of the parameters simulated in HFSS using Maxwell Garnett effective medium theory and average effective thicknesses, no transmittance amplitude is permitted to exceed the bare antenna maximum.

The slits of the metasurface presented in this paper (Figure 1) are approximately 30 microns at most in width. If the interaction of the ∼3 mm wave with the powders at the powder–aperture interface is subject to an increase in the local resonant phenomena due to the powder defects (that have been shown to produce second and third harmonic generation effects due to the nonlinearity of the defect-local field interactions), it is suggested that the metasurface apertures in the area may couple to the strong local fields produced within the defects and may be highly sensitive to local nonlinear phenomenon taking place within them. Such sensitivity to local field enhancements may be producing the anomalous transmittance spectra not predicted by conventional effective medium theories such as Maxwell Garnett [32]. 

## 5. Conclusions

A metasurface for the purpose of fine dielectric powder identification and characterization was designed and built to operate at W-Band. The metasurface device was produced by considering the hallmarks of resonant coupling to the metasurface topology which are optimum for sensing. Four types of easily acquired dielectric powders were individually sifted onto the surface of the metasurface in 200 mg increments, up to 1000 mg, and their transmittance spectra measured. Maxwell Garnett effective medium theory was utilized to estimate the complex dielectric constants of the thin powders, along with an average effective thickness of each powder layer, in order to simulate the resonant response in frequency and transmittance amplitude of thin layers of dielectric powders present on the surface of the device and compare them with experimental data. 

There were notable differences between the experimental spectra and spectra simulated using the Maxwell Garnett effective medium theory. Most glaring were the ability of the metasurface to experimentally produce transmittance spectra of powders whose maxima exceeded that of the bare device (see baking soda and calcium carbonate, Figure 13), and relative frequency shifts which seemed to “jump”, breaking with the expected spectral evolution between successive powder layers (see baking soda, Figure 13), for it was observed in experiment and theory that, more often than not, the relative shift in resonant frequency between successive powder layers decreased as the total powder thickness increased. Furthermore, this anomalous behavior was observed in both runs of the experiment (which were conducted two weeks apart), suggesting that some hitherto unexplained physical phenomena may be the culprit. Nonetheless, a clear distinction between the spectral evolutions of different powders, and the marked similarities between like species suggest that this technique could be used to detect and identify trace amounts of powders on a metasurface at W-band.

In order to study the resonant response of the metasurface device in greater detail, a method by which the metasurface was modeled as a modified capacitive patch array explicitly based upon geometric parameters of the metasurface topology was utilized. It was shown that the proposed equivalent circuit method to extract effective dielectric constants and thicknesses was more successful than the Maxwell Garnett effective medium theory and average effective thicknesses at predicting the relative shifts in resonance values between successive powder layers applied to the metasurface, further reinforcing the claim that this technique could be used for the identification and characterization of thin inhomogeneous powder layers. It has been shown that the proposed metasurface sensing technique serves as a worthy candidate to detect, identify, and characterize extremely thin dielectric powder layers at W-Band.

## Figures and Tables

**Figure 1 sensors-23-00842-f001:**
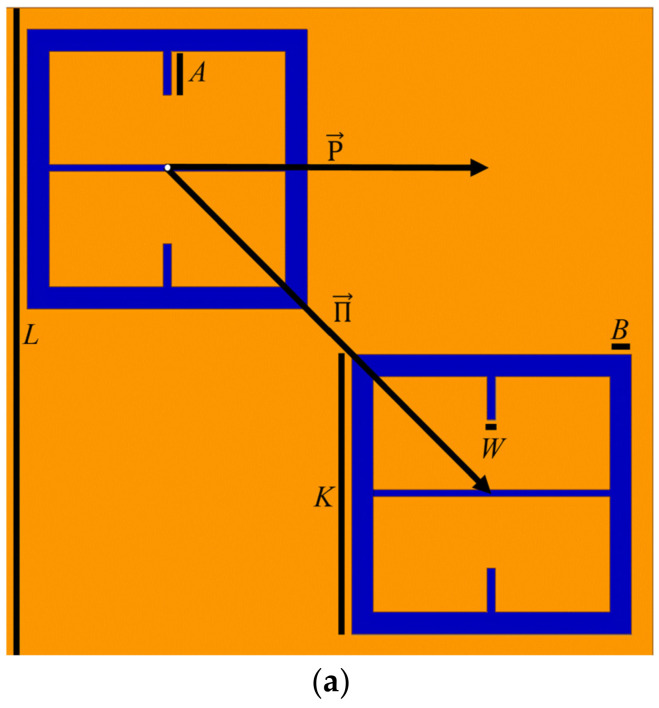
(**a**) The metasurface unit cell which is the focus of this paper (the orange area is a 250 nm thick layer of copper, and the blue is a 400 μm thick silicon wafer beneath it). (**b**) A depiction (side view) of the full metasurface device which shows the many unit cells (**a**) etched in the 250 nm thick layer of copper (orange), atop the 400 μm thick, 4-inch diameter silicon wafer (blue).

**Figure 2 sensors-23-00842-f002:**
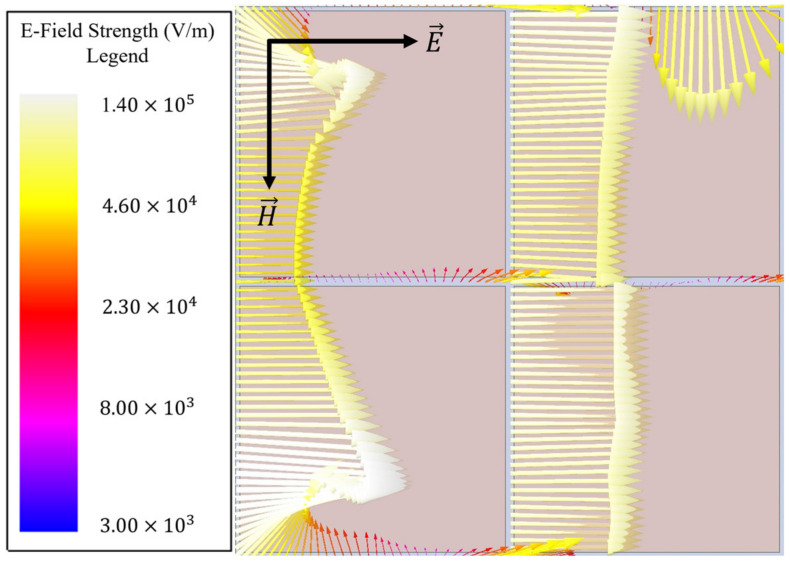
Electric field vectors between aperture slits of a capacitive patch array at 110 GHz, demonstrating resonant coupling of the incident electric field to the array topology.

**Figure 3 sensors-23-00842-f003:**
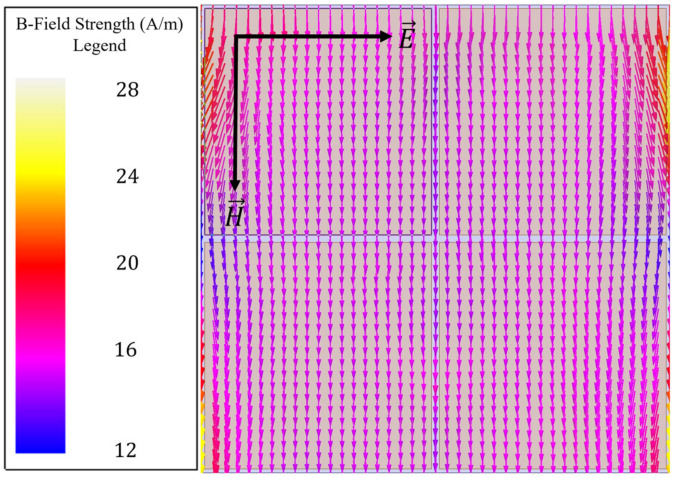
Induced magnetic field vectors over the capacitive patch array at 110 GHz, demonstrating no resonant coupling of the incident magnetic field to the array topology.

**Figure 4 sensors-23-00842-f004:**
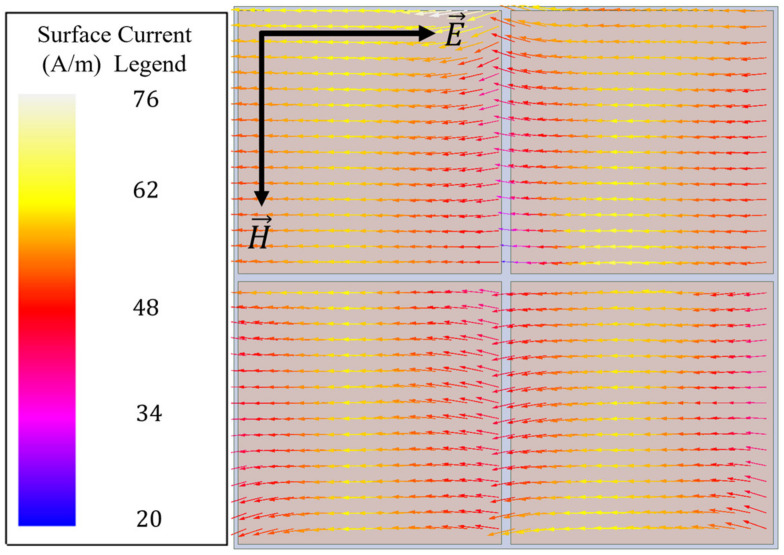
Surface current vectors on the capacitive patch array at 110 GHz.

**Figure 5 sensors-23-00842-f005:**
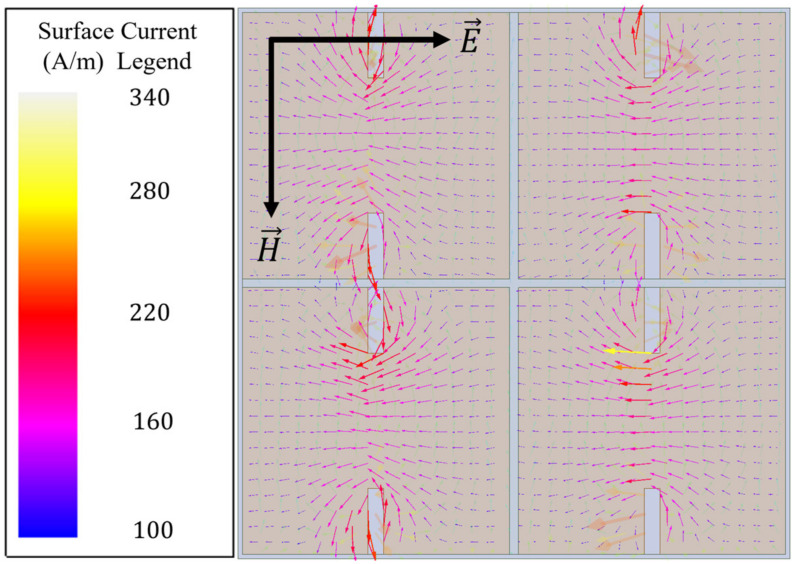
Surface current vectors on the capacitive patch array after perturbative slots were introduced, at 110 GHz.

**Figure 6 sensors-23-00842-f006:**
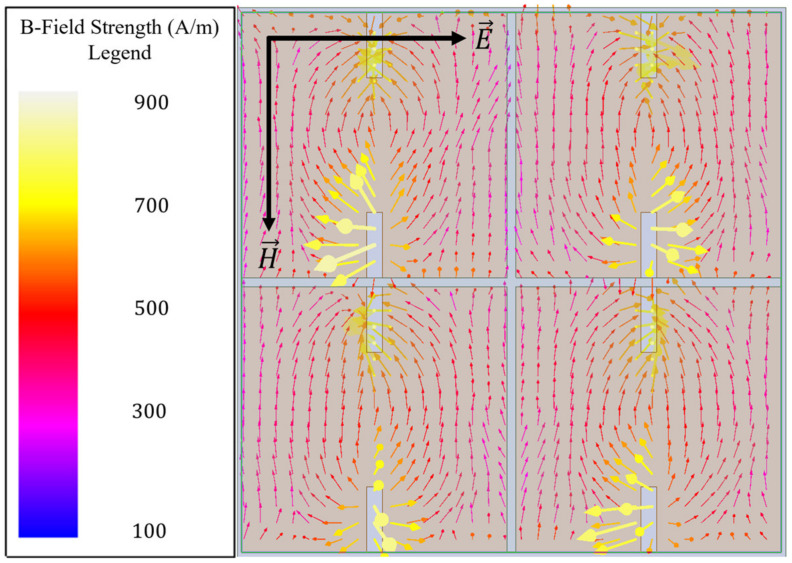
Induced magnetic field vectors directly above the capacitive patch array after perturbative slots were introduced at 110 GHz.

**Figure 7 sensors-23-00842-f007:**
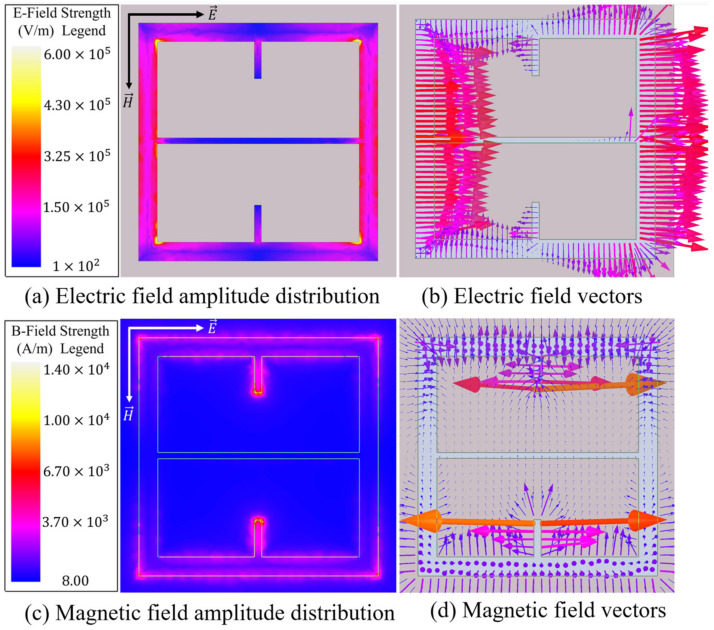
The electric (**a**,**b**) and magnetic (**c**,**d**) fields produced within the aperture slits and 250 nm above the metasurface, respectively, in the device presented in this paper. The electric and magnetic vectors in black (**a**,**b**) and white (**c**,**d**) represent the polarization of the incident radiation on the surface.

**Figure 8 sensors-23-00842-f008:**
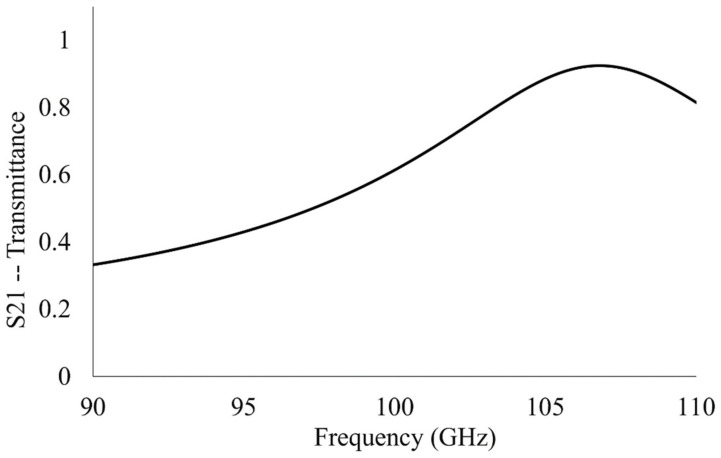
The simulated transmittance spectrum of the metasurface device presented in this paper.

**Figure 9 sensors-23-00842-f009:**
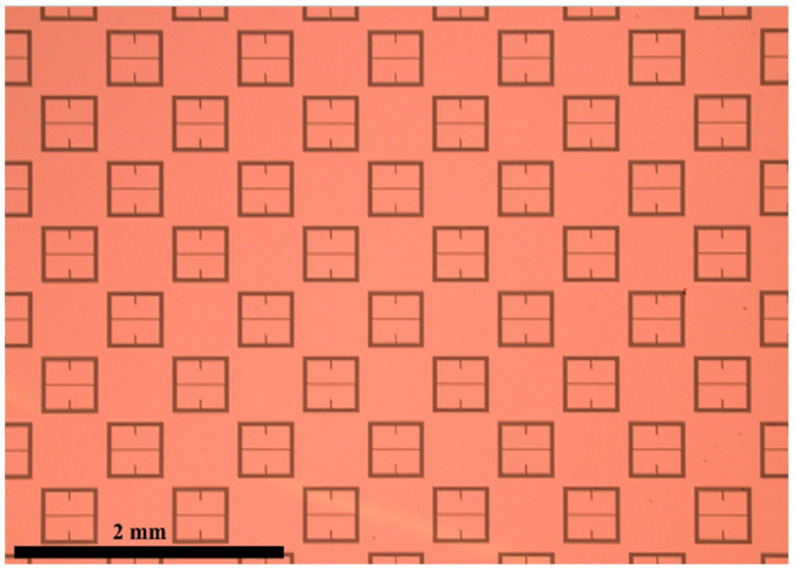
A 2.5× image of a section of the optimized metasurface device manufactured at ETIC and used for fine powder layer sensing in this study.

**Figure 10 sensors-23-00842-f010:**
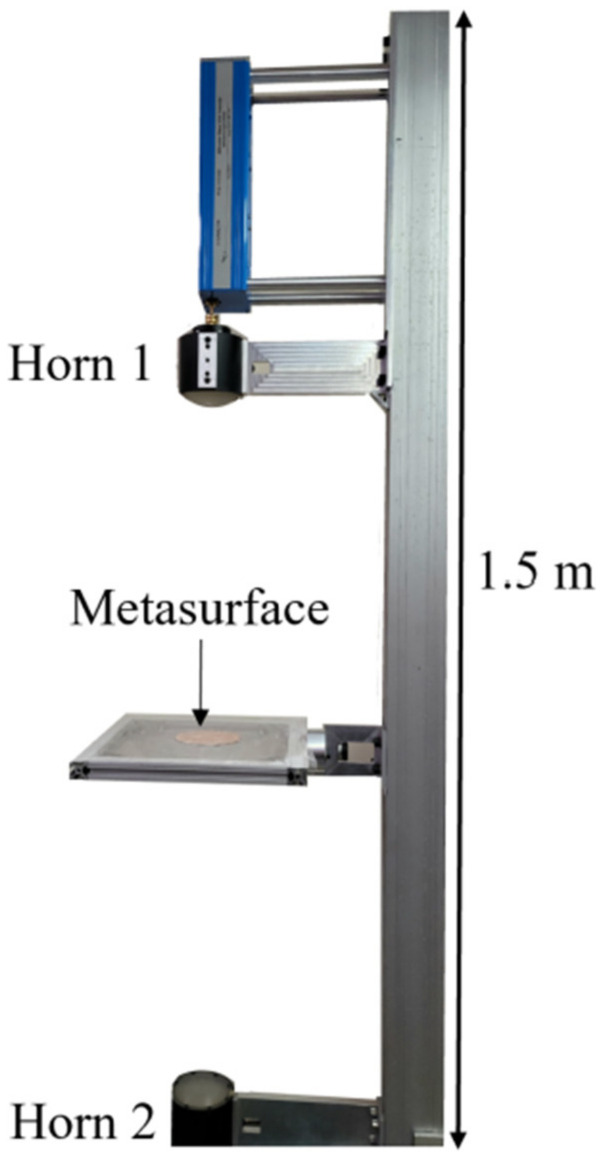
The experimental apparatus utilized to acquire transmittance spectra from the metasurface device.

**Figure 11 sensors-23-00842-f011:**
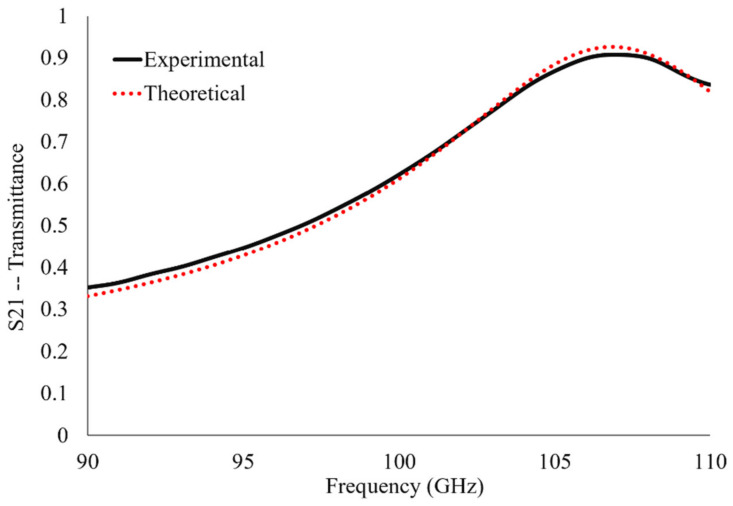
The theoretical and experimentally acquired transmittance spectra of the bare (without powder) metasurface device presented in this paper.

**Figure 12 sensors-23-00842-f012:**
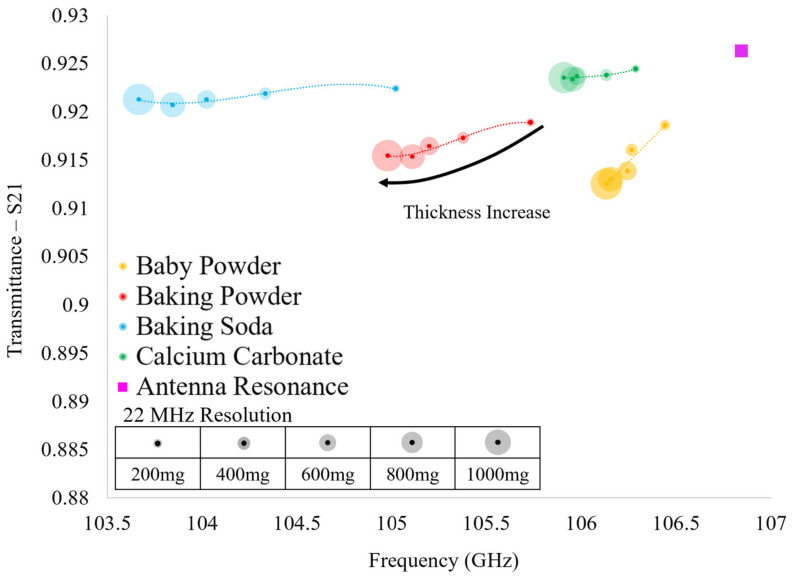
The simulated transmittance spectra of the thin powder layers measured on the metasurface device produced in HFSS using Maxwell Garnett effective medium theory derived complex dielectric constants and average effective thicknesses.

**Figure 13 sensors-23-00842-f013:**
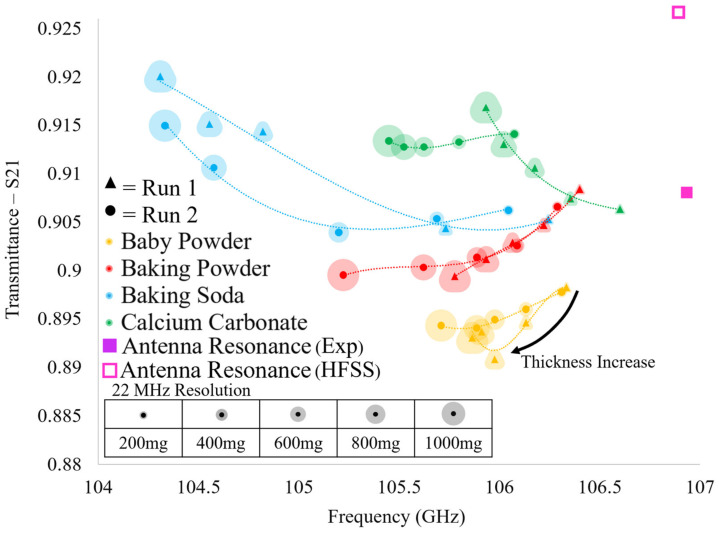
The experimentally acquired transmittance spectra of the thin powder layers measured on the metasurface device over two experimental runs, two weeks apart.

**Figure 14 sensors-23-00842-f014:**
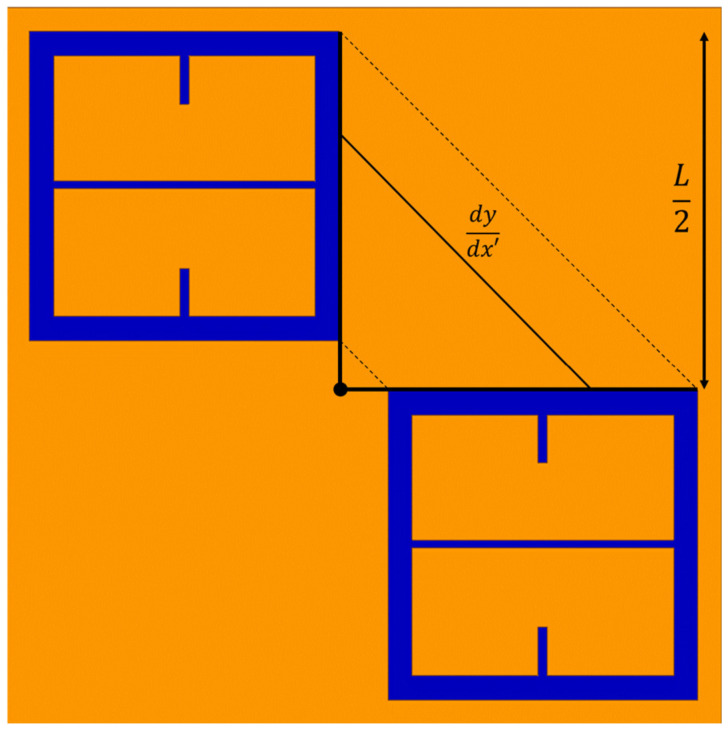
The average inter-patch spacing between the two active unit cell elements of the metasurface, represented as the solid black line labeled $\frac{dy}{dx^{\prime }}$.

**Figure 15 sensors-23-00842-f015:**
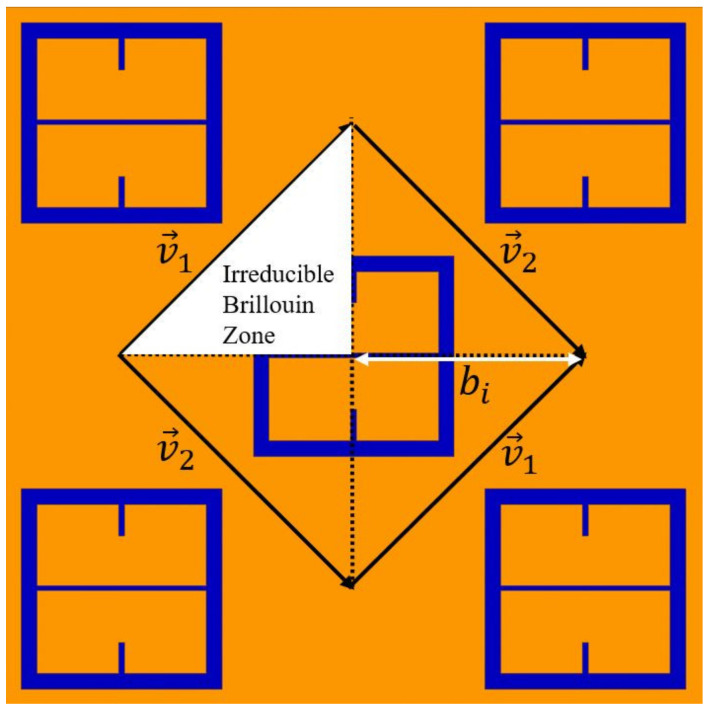
The Irreducible Brillouin Zone on the metasurface lattice. $b_{i}$ represents the length of the interior walls of the Irreducible Brillouin Zone and is also the period (*P*) of the metasurface.

**Figure 16 sensors-23-00842-f016:**
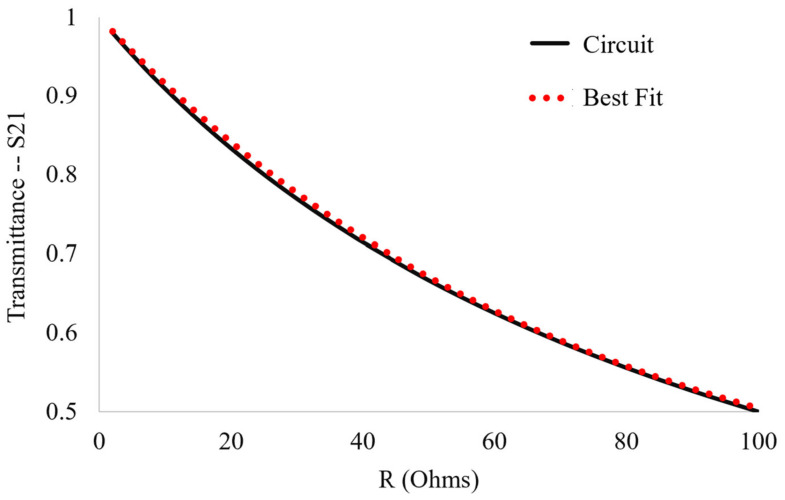
The relationship between the transmittance amplitude and the value of the resistor in the fundamental circuit used to model the resonant response of the metasurface.

**Figure 17 sensors-23-00842-f017:**
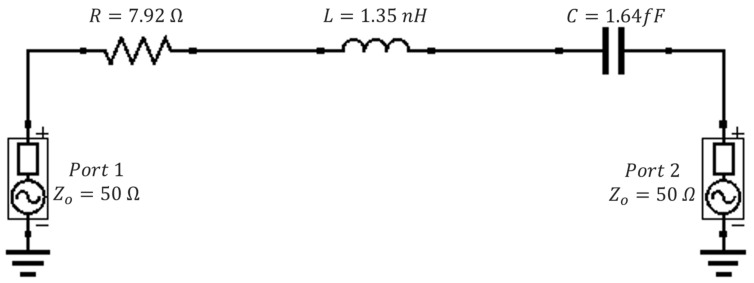
The fundamental equivalent circuit which very closely models the bare metasurface transmittance spectra observed in HFSS.

**Figure 18 sensors-23-00842-f018:**
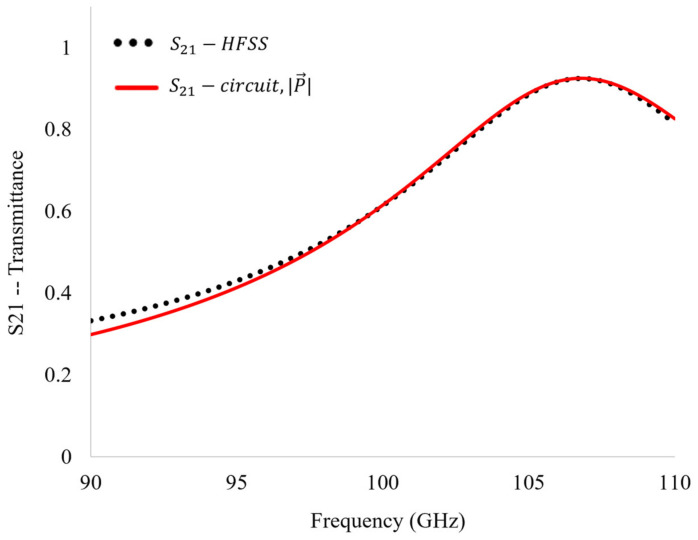
The transmittance spectra produced in QUCS using the fundamental equivalent circuit compared with the transmittance spectra of the bare metasurface produced in HFSS.

**Figure 19 sensors-23-00842-f019:**
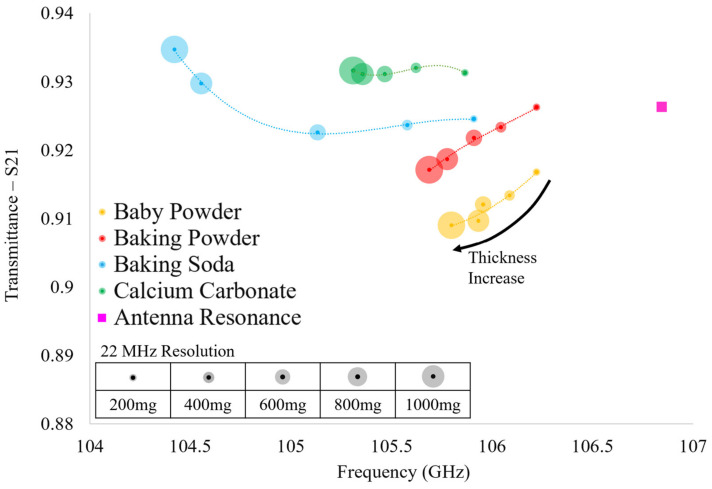
The resonant transmittance spectra of powders simulated in HFSS with the layer thicknesses, permittivities and loss tangents found from the equivalent circuit method, whose values are presented in Table 8 (to be compared with the Run 2 transmittance spectra in Figure 13).

**Figure 20 sensors-23-00842-f020:**
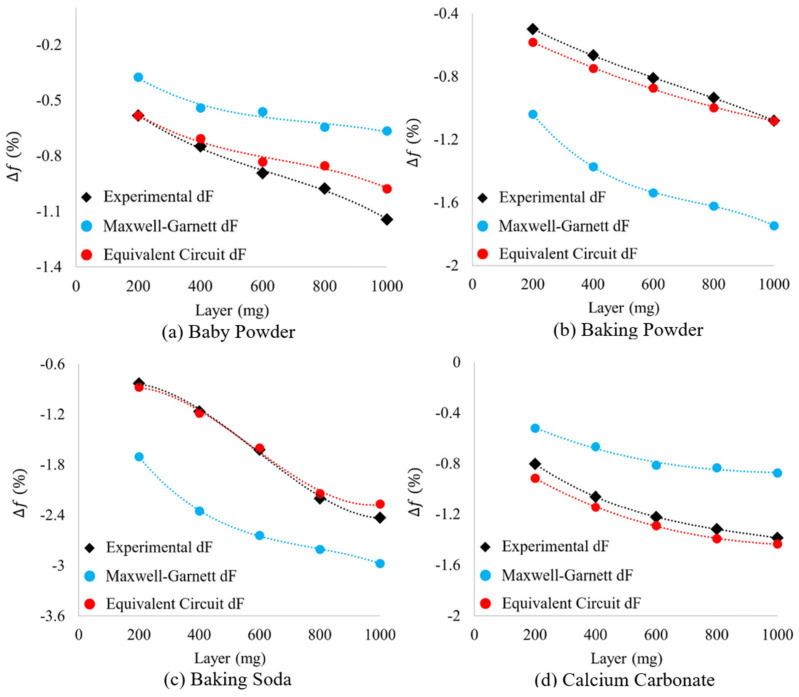
The relative shift in frequencies of experimental and theoretical metasurface powder resonances. The black data points represent the experimentally measured powders, the blue data points represent the changes observed while simulating the powders in HFSS using the Maxwell Garnett effective medium theory and average effective thicknesses, and the red data points represent the relative changes observed for the powders simulated in HFSS with layer thicknesses and complex dielectric constants derived from an equivalent circuit model.

**Figure 21 sensors-23-00842-f021:**
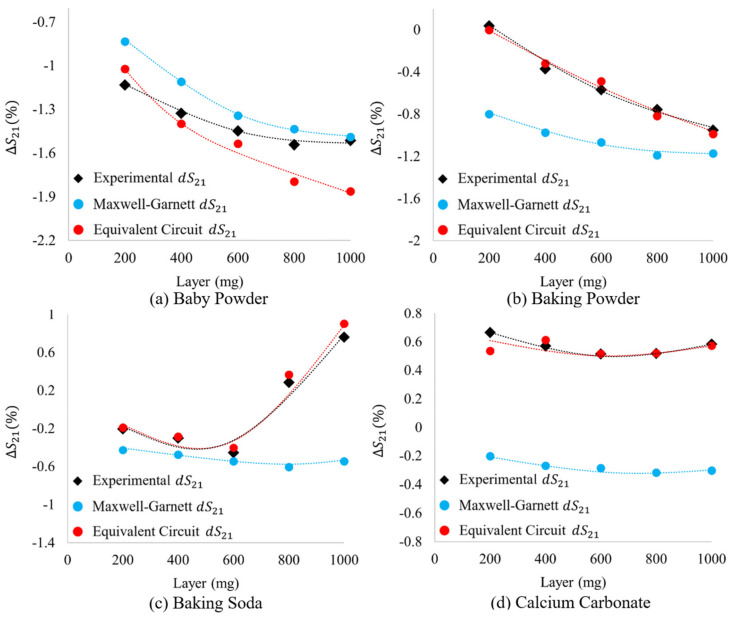
The relative shift in maximum transmittance amplitude of experimental and theoretical metasurface powder resonances. The black data points represent the experimentally measured powders, the blue data points represent the changes observed while simulating the powders in HFSS using the Maxwell Garnett effective medium theory and average effective thicknesses, and the red data points represent the relative changes observed for the powders simulated in HFSS with layer thicknesses and complex dielectric constants derived from an equivalent circuit model.

**Table 1 sensors-23-00842-t001:** Metasurface design parameters corresponding to those depicted in Figure 1.

Symbol	Description	Value (μm)
*L*	Metasurface unit cell length	812
|$\;\vec{\Pi }\;$|	Diagonal lattice period	574
|$\;\vec{P}\;$|	IBZ ^1^ wall length	406
K	Meta-atom major length	350
W	Dipole slot width	10
A	Dipole slot length	55
B	Meta-atom minor width	28

^1^ Irreducible Brillouin Zone.

**Table 2 sensors-23-00842-t002:** Resonant frequency and maximum transmittance amplitude of the bare metasurface (no powders present) measured theoretically (HFSS) and experimentally.

Measurement Type	Resonant Frequency (GHz)	Maximum $S_{21}$
Experimental	106.93	0.908
Theoretical	106.84	0.926

**Table 3 sensors-23-00842-t003:** The total mass of the powders on the metasurface after attempting to sift five 200 mg layers onto it. Run 2 of the experiment was conducted two weeks after Run 1.

Powder	Mass—Run 1 (mg)	Mass—Run 2 (mg)
Baby powder	280	330
Baking powder	480	540
Baking soda	540	640
Calcium carbonate	200	340

**Table 4 sensors-23-00842-t004:** The bulk complex dielectric constants obtained through Fresnel fitting of the reflectance data acquired from the hydraulically pressed powders.

Powder	$\epsilon_{bulk}$
Baby powder	1.54+i0.068
Baking powder	2.33+i0.062
Baking soda	2.71+i0.012
Calcium carbonate	1.88+i0.015

**Table 5 sensors-23-00842-t005:** The average effective thickness ($\;T_{ave}$ ) of each powder layer found using Equation (2) and used in HFSS simulations in order to set a control using Maxwell Garnett effective medium theory.

Powder	$\;T_{ave}$ ( μm)
Baby powder	13
Baking powder	20
Baking soda	17
Calcium carbonate	18

**Table 6 sensors-23-00842-t006:** The estimated fill factors (average fractional amount of air present in each 200 mg powder layer), $\delta_{i}$, used in Maxwell Garnett effective medium theory.

Powder	$\delta_{i}$
Baby powder	0.32
Baking powder	0.48
Baking soda	0.36
Calcium carbonate	0.63

**Table 7 sensors-23-00842-t007:** The complex dielectric properties of the powders obtained by use of Maxwell Garnett effective medium theory.

Powder	$\epsilon_{eff}$
Baby powder	1.35+i0.007
Baking powder	1.62+i0.027
Baking soda	2.00+i0.007
Calcium carbonate	1.29+i0.005

**Table 8 sensors-23-00842-t008:** This table provides all of the physical properties of each powder layer which were derived from the equivalent circuit method (and used in HFSS simulations).

Powder	Layer (mg)	$T_{adj}\;\left(\mu m\right)$	$f_{r}$ (GHz)	$S_{21,r}$	$\epsilon_{eff}^{{}^{\prime }}$	$tan\delta_{eff}$
Baby powder	200	34	106.223	0.9158	1.329	0.0256
	400	49	106.045	0.9140	1.373	0.0307
	600	59	105.890	0.9129	1.412	0.0340
	800	66	105.801	0.9120	1.434	0.0365
	1000	70	105.626	0.9123	1.479	0.0357
Baking powder	200	20	106.312	0.9266	1.307	−0.0030
	400	30	106.134	0.09229	1.351	0.0046
	600	39	105.979	0.9210	1.390	0.0086
	800	49	105.846	0.9193	1.423	0.0128
	1000	60	105.690	0.9175	1.462	0.0174
Baking soda	200	34	105.957	0.9244	1.395	0.0001
	400	42	105.601	0.9235	1.485	0.0011
	600	47	105.113	0.9221	1.610	0.0036
	800	51	104.491	0.9289	1.771	−0.0074
	1000	53	104.247	0.9333	1.835	−0.0136
Calcium carbonate	200	31	105.987	0.9324	1.388	−0.0127
	400	39	105.712	0.9316	1.457	−0.0115
	600	43	105.538	0.9311	1.501	−0.0107
	800	45	105.438	0.9311	1.526	−0.0107
	1000	46	105.367	0.9317	1.546	−0.0117

## Data Availability

Not applicable.

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
