# Peer review of "A Novel Technique for Ultrathin Inhomogeneous Dielectric Powder Layer Sensing Using a W-Band Metasurface"

_sensors, 2023, doi:10.3390/s23020842_

Round 1
Reviewer 1 Report
In the manuscript entitled “A Novel Technique for Ultrathin Inhomogeneous Dielectric Powder Layer Sensing Using a W-Band Metasurface” the authors present a technique using a W-band metasurface for the purpose of transmissive fine powder layer sensing. The authors have claimed that this technique can detect, identify, and characterize inhomogeneous ultrafine powder layers hundreds of times thinner than the incident wavelengths used to sense them. This idea looks innovative, and novel and could be interesting to the major readers of Sensor. Moreover, the manuscript has an incredible similarity of 7% to the previously published literature. However, the following are suggested comments that authors should consider.
1. Check the authors’ name, affiliation (a single email address cannot be used as the affiliation), and numbering (the last author's affiliation and number were missed).
2. Add the findings of this study to the abstract.
3. Abstract should present the significance, finding, conclusion and sometimes recommendations for future work. However, here only the authors’ hypotheses and claims have been presented.
4. How many times have the tests been repeated?
5. There is a lack of compression between the finding of this study and previous literature.
6. Figures should be presented in order as discussed and appear in the text. However, in line 199 figure 11 has been introduced before figure 10 (line 206)
7. What are the possible reasons for the 1.33% error between the experimental and theoretical spectra in figure 11?
Reviewer 2 Report
This paper developed a metasurface working at W-Band for identification and characterization of different powders. The metasurface device was produced by considering the hallmarks of resonant coupling to the metasurface topology which are optimum for sensing. The device showed the capability of distinguishing the four types of powders with different weight. This concept and investigation are interesting and important to the field, this study is data-rich, and the manuscript is easy to follow. Thus, I recommended the manuscript for publication after some minor comments are addressed.
1. How did the authors finalize the unit design, including shapes and parameters? Why choose current parameters, and why these are the best choices?
2. Figure 1 may be confusing for readers to understand the structure of device, I would suggest adding a side view schematic to help readers.
3. What kind of silicon wafer are used here? Will crystal direction or doping have any influence?
4. Authors mentioned that humidity is an important factor to the results. Are the two experiments run in the same humidity and temperature conditions? How will the humidity changes influence the results?
5. How to ensure the distribution of the powers on the device are uniform? How to measure this uniformity?
6. Although the reason for different powers showing different resonant transmittance spectra still needs to be further investigated, and it is difficult to verify there will not exist two powers showing similar spectra, the authors’ current results have shown the device can be used to distinguish these four kinds of powers. I will encourage the authors to further investigate the reason in their future research.
Reviewer 3 Report
In this paper, the authors proposed a novel method for fine powder layer sensing, which is useful during personnel screening processes (i.e., at an airport) and in industrial manufacturing environments where early detection and quantization of harmful airborne particulates can be a matter of security or safety. I think it should be accepted by Sensors when some issues are addressed:
(1) There many grammatical errors, please polish the whole paper.
(2) Is this design functional when the fine powder is not full of layer? Maybe some simulations can be provided.
(3) As shown in Fig. 13, some measured lines are crossed, so how to judge the material and the weight? Please clarify.
